# Relationship between Cyanobacterial Abundance and Physicochemical Variables in the Ebro Basin Reservoirs (Spain)

**Rebeca Pérez-González** [1,*] , **Xavier Sòria-Perpinyà** [2] , **Juan Soria** [1] , **Maria D. Sendra** [1] **and Eduardo Vicente** [1]

1   Cavanilles Institute of Biodiversity and Evolutionary Biology (ICBiBE), University of Valencia, 46980 Paterna, Spain
2   Image Processing Laboratory (IPL), University of Valencia, 46980 Paterna, Spain
*   Correspondence: rebeca.perez@uv.es

**Abstract:** One of the main problems arising in inland waterbodies is nutrient enrichment that accelerates eutrophication, causing massive cyanobacteria blooms and degrading aquatic ecosystems. This study focused on physical/chemical factors that affect cyanobacteria of 30 reservoirs in the Ebro River basin within the Iberian Peninsula of northeastern Spain. The abundance of cyanobacteria was assessed as total cell number, total biovolume, and the indicator pigment, total phycocyanin (PC). In addition, empirical measurements for PC were compared to PC estimated from remote sensing. Variables assessed for correlation with cyanobacteria abundance included temperature, pH, light availability inferred from Secchi depth, water residence time, total nitrogen, dissolved inorganic nitrogen, total phosphorus, soluble reactive phosphorus, silica, and total phytoplankton biomass as chlorophyll *a*. These variables were also assessed with a multi-statistical principal component analysis for relationships with cyanobacteria abundance. Cyanobacteria cell number and biovolume were positively correlated with temperature, total nitrogen, total phosphorus, and water residence time, and negatively correlated with silica. High PC concentrations were documented in the reservoirs, and satellite images from remote sensing showed the PC spatial distribution and heterogeneity in the reservoirs. The PCA results show that some variables, such as nitrogen and phosphorus, are closely related to the abundance of cyanobacteria, while other variables such as silica do not show a clear relationship. This study contributes to the knowledge base about inland waterbodies from a physical/chemical perspective, which had not been done before in the Ebro Basin, including the application of analytic tools such as remote sensing.

**Keywords:** cyanobacterial biovolume; harmful cyanobacteria blooms; phycocyanin; remote sensing; reservoirs

## 1. Introduction

Inland waterbodies are subject to significant pressures as a result of human activities. On the one hand, there is a growing need for water supplies due to population growth, while on the other, there is increasing risk of water contamination occurring, in most cases, from point and nonpoint source discharges without adequate treatment. As a consequence, the ecological status of waterbodies is worsening [1], resulting in the increasing occurrence of algal blooms, presenting a great risk to freshwater flora and fauna, as well as to human users.

The proliferation of cyanobacteria is commonly known as outbreaks or cyanobacterial harmful algal blooms (CHABs). Such blooms typically are fueled by nutrient contamination (phosphorus, P, and nitrogen, N) from agriculture, septic systems, sewage, and other human sources [2]. These blooms can disrupt ecosystems, estuarine, and coastal marine ecosystems, and are becoming increasingly common [1,3]. Not only do they promote water quality degradation, but they can also cause significant economic losses, because, in cases where the waters are used for potable purposes, purification treatments are expensive.

Cyanobacteria blooms refer here to massive biomass production, often characterized by the blue-greenish or other discoloration of the water [4]. These high-biomass blooms used to occur mainly in warm periods [5], especially near the end of summer, but climate change is lengthening the "bloom season" in many temperate environments [6]. Water discoloration can be accompanied by foam, scums, and vomit-like odors and potent toxins that threaten recreational activities as well as the consumption of such water [3]. Furthermore, such blooms can also cause hypoxia and anoxia from high algal respiration at night, and from the decomposition of dying/dead algae which, in turn, can cause the death of beneficial aquatic organisms [3,4].

For cyanobacterial blooms, some favorable conditions can naturally be present, especially nitrogen (N) and phosphorus (P) enrichment that is greatly exacerbated by human-related inputs [6]. The hydrological regime is also important, as rivers and lakes differ markedly in water circulation and residence times. The bottom sediments can accumulate nutrients and then release them as conditions become anoxic, stimulating cyanobacteria and other primary producers [6].

Cyanobacteria are "on the border" of prokaryotes and photosynthetic eukaryotes, as they possess characteristics associated with bacteria, such as the absence of organelles such as a nucleus and chloroplasts, as well as the oxygenic photosynthesis of eukaryotic algae. They mostly function ecologically as primary producers. Most cyanobacteria exhibit the following combination of characteristics—certain morphological and cellular traits, aerobic metabolism, chlorophyll *a* and light antennae phycobilin pigments, and the ability of many planktonic taxa to control their vertical position in a pressure-dependable water column [7].

Some cyanobacteria can use an energetically costly process to fix atmospheric $N_2$ into inorganic N (ammonia), and all taxa are able to obtain N from water through the assimilation of $NH_4^+$, $NO_3^-$, or various organic N forms (proteins, amino acids…) [8]. Both P and N may be limiting or co-limiting resources for phytoplankton growth [6], and both can be critical parameters in water quality due to their influence on eutrophication [9]. Cyanobacteria have a high P storage capacity, and many cyanobacteria have high optima for both P and N [10].

The objective of this research was to analyze summer cyanobacterial abundance in 30 reservoirs within a north temperate watershed, and potential influencing variables such as temperature, pH, water residence time, and P and N forms. This study contributes a novel comparison of data from in situ sampling versus remote sensing as a tool for monitoring, mitigation, and improved forecasting of high-biomass CHABs. Diverse cyanobacterial toxins (cyanotoxins) are among the potent toxins known, affecting beneficial wildlife, aquatic life, and humans [3]. The reservoirs included in this study were also evaluated based on their status as drinking-source waters, using World Health Organization (WHO) guidance for cyanobacterial abundance [11].

## 2. Materials and Methods

### 2.1. Study Area

The Ebro River basin is in the northeast region of the Iberian Peninsula (Figure 1). It is the largest river basin in Spain, with a total surface area of 85.534 km$^2$ [12]. The area is mainly characterized by a Mediterranean climate, with oceanic influences in the northwest and a more continental climate in the central peninsula.

This study was conducted from 2015 through 2019 in the summer season, defined as the period from June through September. Ebro basin has a Mediterranean climate with oceanic influences in the northwest and continental or mountain features in the interior. Precipitation is concentrated in the mountainous periphery, mainly in the Pyrenees, where it exceeds 1000 mm/year, while in the central valley it does not exceed 400 mm/year, and conditions are semi-arid. The 30 reservoirs are briefly described in Table 1, and further details are given in Appendix A—Table A1. For some analyses, the reservoirs were grouped into 7 of 13 types according to the Water Framework Directive [13] based on climatic conditions (mean annual temperature, humidity) and watershed lithology

(calcareous versus siliceous) and area (categories: <1000, >1000, between 1000 and 25,000, and more than 25,000 km²) (Appendix A—Table A2). It should be noted that reservoirs belonging to types 1 and 13 were combined into type 1, since both types occur in high mountainous regions (one calcareous and the other siliceous). Two reservoirs were type 1 (Monomictic, Siliceous, Medium Humidity), 20 were type 7 (Monomictic, Calcareous, Medium Humidity), six were type 9 (Monomictic, Calcareous, Medium Humidity), 12 were type 10 (Monomictic, Calcareous, Low Humidity), 12 were type 11 (Monomictic, Calcareous, Low Humidity), four were type 12 (Monomictic, Calcareous, Low Humidity), and four were type 13 (Dimictic, High Humidity).

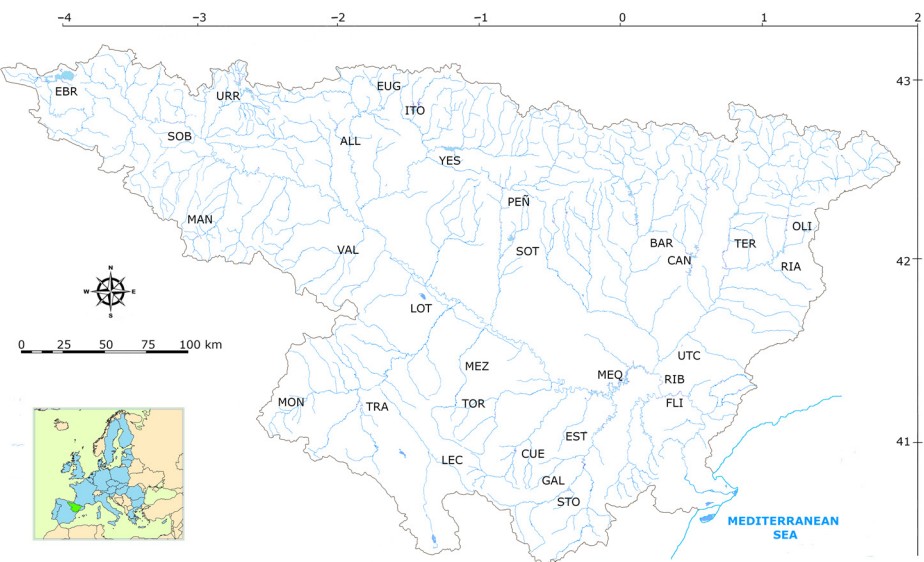

**Figure 1.** Map of the Ebro basin, indicating geographical coordinates and the locations of the 30 sampled reservoirs. Names indicated by abbreviations are given in Appendix A.

**Table 1.** Reservoirs description. Position according to geographical coordinates (Latitude and Longitude).

| Name | Position | | Depth m (max) | Volume ×10⁶ m³ | Elevation m.a.s.l | Res. Time (Years) | Climate |
|---|---|---|---|---|---|---|---|
| | Lat. | Lon. | | | | | |
| **Alloz** | 42.70 | −1.92 | 60 | 65 | 468 | 0.48 | Cfa |
| **Barasona** | 42.14 | 0.33 | 66 | 85 | 448 | 0.24 | Cfa |
| **Canelles** | 42.03 | 0.65 | 150 | 201 | 506 | 0.00 | Cfb |
| **C. Foradada** | 40.97 | −0.69 | 65 | 22 | 580 | 0.65 | Bsk |
| **Ebro** | 42.97 | −4.07 | 34 | 540 | 838 | 1.55 | Cfb |
| **Est. Alcañiz** | 41.06 | −0.18 | 15 | 7 | 342 | 0.14 | BSk |
| **Eugui** | 42.97 | −1.51 | 43 | 21 | 628 | 0.18 | Cfb |
| **Flix** | 41.23 | 0.53 | 26 | 11 | 41 | 0.01 | BSk |
| **Gallipuén** | 40.87 | −0.41 | 36 | 4 | 694 | 0.71 | Cfb |
| **Itoiz** | 42.48 | −1.21 | 107 | 418 | 573 | 0.57 | Cfb |
| **Lechago** | 40.96 | −1.30 | 18.5 | 7 | 891 | - | Csa |
| **Loteta** | 41.82 | −1.32 | 34 | 100 | 288 | 3.51 | BSk |
| **Mansilla** | 42.16 | −2.91 | 70 | 68 | 930 | 0.09 | Cfa |
| **Mezalocha** | 41.42 | −1.07 | 45 | 4 | 473 | 1.17 | Cfa |

**Table 1.** *Cont.*

| Name | Position | | Depth m (max) | Volume ×10$^6$ m$^3$ | Elevation m.a.s.l | Res. Time (Years) | Climate |
|---|---|---|---|---|---|---|---|
| | Lat. | Lon. | | | | | |
| Mequinenza | 41.22 | 16.33 | 79 | 1534 | 106 | 0.13 | Csa |
| Moneva | 41.17 | −0.83 | 45 | 8 | 615 | 0.95 | Cfb |
| Oliana | 42.12 | 1.3 | 102 | 84 | 519 | 0.08 | Cfa |
| Peña | 40.82 | 0.13 | 61 | 18 | 561 | - | Bsk |
| Rialb | 41.97 | 1.23 | 99 | 402 | 430 | 0.36 | Cfa |
| Ribarroja | 41.33 | 0.36 | 60 | 207 | 70 | 0.03 | Csb |
| Santolea | 40.77 | −0.31 | 44 | 48 | 596 | 0.60 | Csa |
| Sobrón | 42.76 | −3.15 | 39 | 20 | 511 | 0.06 | Cfb |
| Sotonera | 42.11 | −0.68 | 31 | 189 | 417 | 0.58 | Cfa |
| Terradets | 42.05 | 0.88 | 47 | 33 | 372 | 0.04 | Cfa |
| Torcas | 41.29 | −1.08 | 41 | 7 | 624 | 0.27 | Cfa |
| Tranquera | 41.24 | −1.78 | 81 | 84 | 684 | 0.68 | BSk |
| Urrúnaga | 42.98 | −2.65 | 31 | 72 | 547 | 0.31 | Csb |
| Utchesa | 41.51 | 0.52 | 16.6 | 4 | 147 | - | Cfb |
| El Val | 42.61 | −1.78 | 50 | 24 | 620 | 0.42 | Cfa |
| Yesa | 42.61 | −1.18 | 60.7 | 447 | 488 | 0.23 | Cfb |

### 2.2. In Situ Sampling

Samples were collected by boat during summers from 2015 to 2019 (except for PC, which was sampled from 2016 to 2019) under favorable weather conditions when wind speed was less than 10 km/h. Higher wind speeds would have interfered with images captured by the satellites. A Garmin GPS device was used to locate each sampling station.

Water depth was measured using a hand-held sonar device (DIVE-SCAN, Professional sonar system). Water transparency was measured using a Secchi disk (Secchi Disk Depth, SDD) following [14]. Temperature and pH were measured using a pH meter (Multi 350i/SET, WTW).

For chemical and biological variables, integrated water samples of water photic zone according to DMA were collected in a Ruttner hydrographic sampler, then transferred to bottles (Nalgene bottles for total suspended matter and chlorophyll *a*, glass bottles and Falcon tubes for phosphorus and nitrogen, and topaz glass bottles for phytoplankton) and transported to the laboratory in darkness on ice. Samples were held under appropriate conditions until analysis.

Water residence time (RT) was calculated as the ratio between the average daily outflow of the reservoir and the estimated total reservoir volume, following [15].

$$RT = V/I \tag{1}$$

where V: water volume in m$^3$ and I: outflow in m$^3$ day$^{-1}$.

Daily water volume and outflow data from reservoirs was obtained from Ebro Basin Authority (Spanish Government) [12].

### 2.3. Laboratory Analysis

2.3.1. Chemical Variables

Total P was determined following the Murphy and Riley method [16], with practical quantification limit 0.01 mg P L$^{-1}$. Soluble reactive phosphorus (SRP) was analyzed following Murphy and Riley [16], with a detection limit and limit of quantification of 0.1 µg P L$^{-1}$.

Nitrate+nitrite, hereafter referred to as nitrate, was measured spectrophotometrically using method APHA 4500-$NO_3$/E/4500-$NO_2$ B, with a practical quantification limit of 0.01 mg N/L and a detection limit of 0.0003 mg N/L [17]. Total N was determined using method APHA 4500-N C [17] after N moieties were oxidized to nitrate via digestion with persulfate under alkaline conditions. Silica ($SiO_2$) was measured following method APHA 4500-$SiO_2$C, with a practical quantitation limit of 0.1 mg $SiO_2$ $L^{-1}$ [17].

### 2.3.2. Phytoplankton

Samples for chlorophyll *a* analysis were held in darkness at 4 °C until analysis following Shoaf and Lium [18]. Chlorophyll *a* concentration was calculated according to Equation (2) of Jeffrey and Humphrey [19]:

$$Ca = (11.85 \cdot A664) - (1.54 \cdot A647) - (0.08 \cdot A630) \tag{2}$$

$$Chl\ a\ (mg/L) = [Ca \cdot V_1]/[L \cdot V_2]$$

where A is the absorbance coefficient for different wavelengths, $V_1$ is the extract volume in liters, $V_2$ is sample volume in $m^3$, and L is the light path of the cuvette in cm.

To determine the total phycocyanin (PC) concentration, water samples were analyzed following [20] with a Hitachi Fluorescence Spectrophotometer FL-7000 (Hitachi Ltd., Tokyo, Japan). A *Spirulina.* standard (40% purity; Sigma-Aldrich CAS 11016-15-2, St. Louis, MO, USA) was used for calibration.

Samples for phytoplankton assemblage analysis were preserved immediately in the field with acidic Lugol's solution [21] and held in darkness at 4 °C until analysis within 2–4 weeks. Samples were analyzed for total cyanobacterial cell number and biovolume with the Utermöhl method [22] using a Nikon Eclipse Ti-U inverted microscope (Nikon Corp., Tokyo, Japan).

Cyanobacterial cells were counted until a representative percentage of species was obtained, in order to reduce the standard deviation as much as possible. With some exceptions, a minimum of 100 fields per chamber were counted at 60× and 100× to obtain the cell number (density) per milliliter (cells $mL^{-1}$), and to achieve acceptable precision as stipulated in [19]. Biovolume was estimated by multiplying the cell number of each species by the average volume of each cell [23]. Values in $\mu m^3$ $mL^{-1}$ were transformed into $mm^3$ $L^{-1}$ for statistical description and analysis.

### 2.4. Data Processing

Data were grouped by reservoir type as defined in the Water Framework Directive [13]. Linear regression analyses were first conducted for cyanobacteria abundance (cell number, biovolume, phycocyanin) versus temperature, pH, water residence time, and nutrient concentration (TN, inorganic N, TP, phosphate, silica). Significant differences between reservoir types were then assessed on log-transformed data using two-way ANOVA, Tukey's pairwise significant comparison, and Pearson's correlations [24].

Principal component analysis (PCA, [25]) was carried out on log-transformed data to evaluate factors influencing cyanobacterial biovolume and to examine factor relationship. Reservoirs with minimal cyanobacteria, inferred from negligible PC concentration, were excluded from this analysis.

### 2.5. WHO Classification of Water Bodies for Human Health Risk from Cyanobacteria

The WHO [11] has established guidelines for drinking water quality with respect to cyanobacteria and their toxins, based on the assumption that the waters are adequately monitored. The WHO guidance tacitly recognizes the fact that cyanotoxins, of most interest from a human health perspective, are expensive to measure and not analyzed by many laboratories. Thus, despite the fact that there is an undependable relationship between cyanobacterial abundance and cyanotoxin concentration [3,26], the WHO guidance is based

on cyanobacterial abundance in terms of cell number, biomass (biovolume), and chlorophyll *a* content, as well as the cyanobacteria signature total phycocyanin pigments (Table 2).

**Table 2.** WHO classification of waterbody risk to human health based on cyanobacterial abundance as total cell number (density), total biovolume, and chlorophyll *a* following Chorus and Welker [3], and PC following Brient et al. [27].

| Drinking Water | Bath Water | Density (cel./mL) | Biovolume (mm$^3$/L) | Chlorophyll *a* (µg/L) | PC (µg/L) |
|---|---|---|---|---|---|
| Surveillance level | | 200 | 0.02 | 0.1 | <0.1 |
| Alert level I | | 2000 | 0.2 | 1.0 | 4 |
| | Guidance level I | 20,000 | 2 | 10 | 30 ± 2 |
| Alert level II | Guidance level II | 100,000 | 10 | 50 | 90 ± 2 |

*2.6. Remote Sensing*

Remote sensing provides information about the Earth's surface from sensors on satellites by means of a continuous electromagnetic interaction. This satellite imagery can be used to detect algal blooms in near-real time [28–30]. Among the missions that have been optimized for water quality assessment are the Sentinel missions, especially Sentinel-2 and Sentinel-3, supported by the European Space Agency [27]. The detected suspended microalgal biomass and density are directly proportional to the most abundant algal pigments such as chlorophyll *a*, which is present in all organisms with oxygenic photosynthesis, and PCs, which are assumed to be indicative of cyanobacteria [7,31].

Images corresponding to the field sampling dates were used from the Sentinel-2 satellite, which offers the best spatial resolution for medium-sized waterbodies characteristic of the Iberian Peninsula reservoirs. The images were downloaded from the ESA server, Open Access Hub. Once downloaded in L1C format, they were processed using the Sentinel Platform Application (SNAP) software (Brockmann Consult Gmbh, Hamburg, Germany), and resampled so that all bands had the same spatial resolution. The areas of interest were cropped, and the atmospheric correction procedure Case 2 Regional Coast Color Extreme Complex water (C2RCC-C2X) was applied, a correction that has been validated for similar case studies in turbid waters. This type of atmospheric correction (C2RCC-C2X) was developed for turbid waters. The atmospheric correction methods employed reflectance and radiance data obtained via simulation using radiative transfer models [29].

Phycocyanin concentrations were estimated using a formula from Pérez-González et al. [31], after calibration and subsequent validation of field data from the study region and satellite reflectance. The Sentinel-2 sensor does not have a band with a specific wavelength corresponding to that of phycocyanin [29]. The following equation was used to relate two of the bands present in the Sentinel-2 reflectance spectrum, so that PC concentrations could be estimated:

$$[PC] = 24{,}665 \times (R_{705}/R_{665})^{3.4607} \tag{3}$$

where [PC] is the estimated PC concentration (µg L$^{-1}$), R705 is the atmospherically corrected angle-dependent remote sensing reflectance (RRS) of the band at 705 nm, and R665 is the RRS of the band at 665 nm (R$^2$ = 0.7, RMSE = 8.1 µg L$^{-1}$, and RRMSE = 18.91%).

**3. Results**

In total, 50 samples were taken during summer from 30 reservoirs, with variable numbers of samples being collected from each reservoir in different years (Appendix A—Table A1).

*3.1. Physical Conditions*

Samples were collected during summer periods, and temperatures were similar between reservoirs within a type (Figure 2A; also see Appendix A, Table A2). Type 1 reservoirs had the minimum temperatures, whereas maximum temperatures occurred in type 12 reser-

voirs. Temperatures in type 1 reservoirs were significantly lower than in the other reservoir types. (Figure 2A). Other reservoir types were somewhat similar in temperature regime; however, they follow a similar distribution: the average values for reservoir types 10 and 11 were above reservoir types 7 and 9.

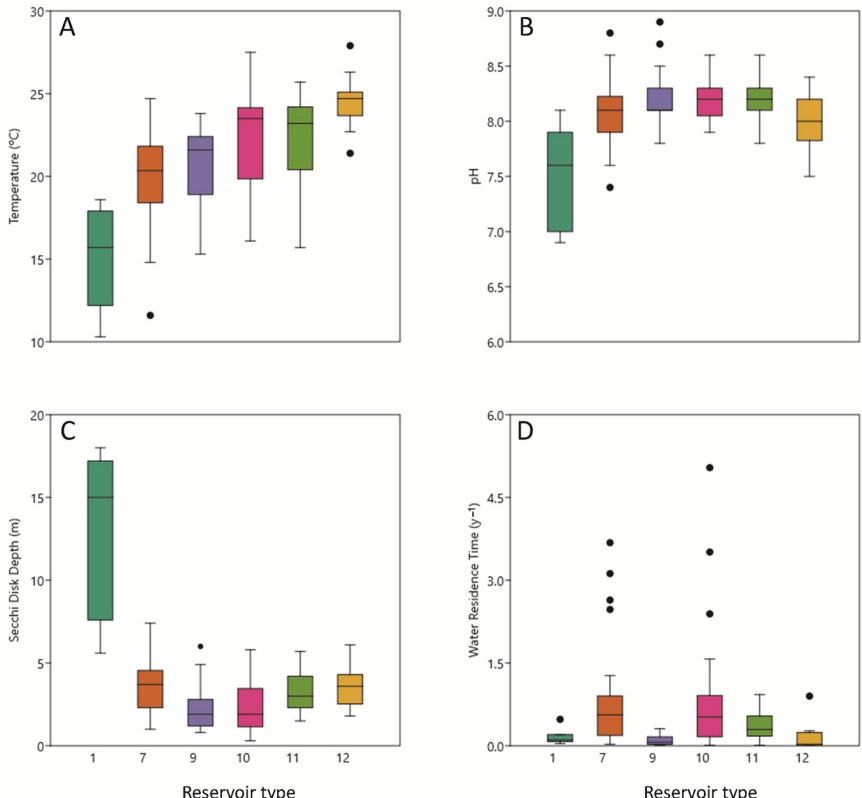

**Figure 2.** Box plots as a function of reservoir type, including (**A**) surface water temperature (°C), (**B**) surface water pH, and (**C**) Secchi depth; and (**D**) water residence time. Each box bounds the interquartile range (IQR; 25–75 percentile), the horizontal line inside each box indicates the median, and "whiskers" (error bars) indicate the 90th and 10th percentiles above and below, respectively. Dots indicate outliers, and letters along the top of each graph indicate significant differences (ANOVA statistical test, $p < 0.05$) between reservoir types.

Reservoir surface water pH values typically indicated alkaline conditions (Figure 2B). The type 1 reservoirs had significantly lower pH values than in other reservoir types. The reservoirs classified as belonging to type 1 presented pH values that were statistically significantly different from those of reservoirs belonging to other types.

The Secchi disk values were similar among reservoirs of the same typology; however, the type 12 reservoirs differ slightly from the rest due to their size and great depth. Furthermore, it is worth noting that more eutrophic reservoirs will have lower transparency compared to mesotrophic or oligotrophic ones (Figure 2C).

Water residence time was <1 y for 83% (25 of 30) of the reservoirs (Figure 2D; mean 0.62 $y^{-1}$), which is characteristic of "annual" reservoirs, that is, reservoirs that fill during the wet period and empty during the dry period. The shortest water residence times were estimated for reservoir types 1, 9 and 12 (0.03 to 1.5 $y^{-1}$); longest residence times were estimated for reservoir types 7 and 10.

### 3.2. Chemical Conditions

According to the trophic state categories established by the OECD (Organization for Economic Cooperation and Development), TP levels < 10 µg $L^{-1}$ correspond to an oligotrophic state, values between 10 and 35 µg $L^{-1}$ correspond to a mesotrophic state,

and values above 35 µg L$^{-1}$ correspond to a eutrophic state (Figure 3A). Maximum TP (>140 µg L$^{-1}$) occurred in the Utchesa (2016) and Moneva (2017) reservoirs, which also sustained high cyanobacterial biomass (total biovolume at $2.15 \times 10^5$ µg m$^{-3}$ and $3.48 \times 10^5$ µg m$^{-3}$, respectively). Significant differences in TP were not discerned among reservoir types.

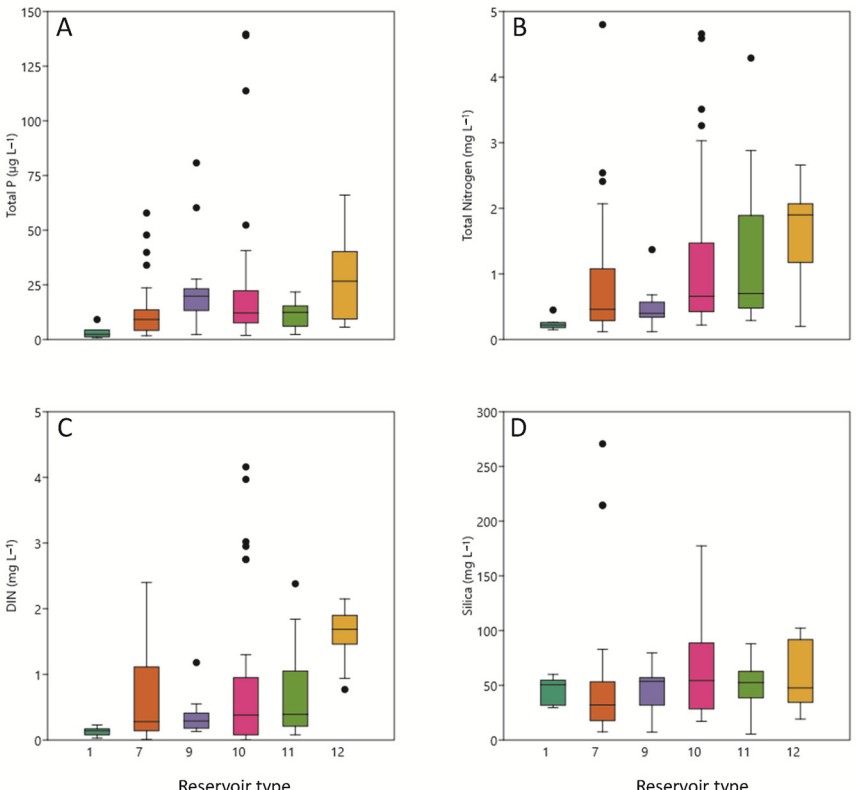

**Figure 3.** Box plots of variables as a function of reservoir type, including surface water concentrations of (**A**) TP, (**B**) TN, (**C**) dissolved inorganic N (DIN), and (**D**) silica. Data are presented as in Figure 2.

Total N was maximal at ~4.79 mg L$^{-1}$ in 2017 in El Val reservoir during summer 2017 and coincided with high cyanobacterial biovolume ($3.22 \times 10^5$ µg m$^{-3}$). Dissolved inorganic nitrogen (DIN) was generally abundant and was not significantly correlated with total cyanobacterial biovolume. Type 12 reservoirs had significantly higher concentrations of both TN and DIN than the other reservoir types (Figure 3B,C).

As would be expected [6], silica exhibited a different distribution from that of N and P (Figure 3D). Significant differences were not discerned among reservoir types. Reservoirs with higher silica concentrations tended to have fewer cyanobacteria. For example, maximal silica was measured in Lechago reservoir during summer 2015 (16.24 mg L$^{-1}$), and cyanobacterial PC concentrations were negligible.

### 3.3. Phytoplankton Pigments

The biovolume and total density of cyanobacteria in the studied reservoirs were obtained on the basis of phytoplankton analysis. Cyanobacteria were found in 97 samples and were absent from 39 samples (Figure 4).

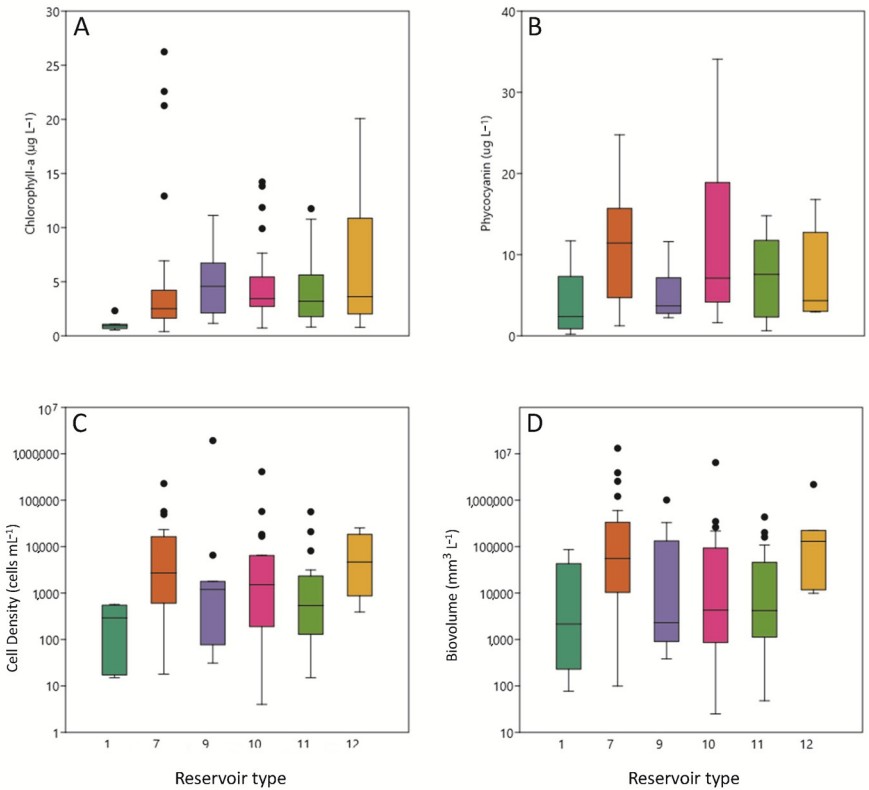

**Figure 4.** Box plots showing variables as a function of reservoir type, including (**A**) chlorophyll *a* (chl *a*), indicator of total phytoplankton biomass (**B**) phycocyanin (PC), indicator of cyanobacterial biomass; (**C**) Total cyanobacteria cells; and (**D**) total cyanobacteria biovolume. Data are presented as in Figure 2; note that significant differences among reservoir types were not discerned for total cells or total biovolume.

Summer PC concentrations (2016–2019) were similar among reservoirs except that type 7 reservoirs had significantly less PC than type 10 reservoirs (Figure 4C). Phycocyanin is a variable closely related to cyanobacteria biovolume, because it constitutes a characteristic pigment of cyanobacteria.

Summer chlorophyll *a* concentrations were quite similar (Figure 4A), except for in type 1 reservoirs.

Cyanobacterial biovolume and cell number (Figure 4C,D) were very similar. In 2015, the reservoir with the highest presence of cyanobacteria was Guiamets, in 2016 it was Urrúnaga, in 2017 it was Ortigosa, in 2018 it was El Val, and finally in 2019 it was Cueva Foradada, all of which contained large amounts of cyanobacteria over the five-year sampling period. The highest biovolume of cyanobacteria is recorded was in 2017 in Ortigosa, among the other reservoirs and years. Furthermore, there were other reservoirs in which the absence of cyanobacteria was maintained over the years, such as Mezalocha, Ciurana, and Terradets.

### 3.4. Phytoplankton Assemblages

The reservoirs did not differ statistically in total cyanobacteria cell number or total cyanobacteria biovolume (Figure 4C,D). Maximum cyanobacteria cell number was measured in surface waters of the Cueva Foradada Reservoir in 2019 ($1.93 \times 10^6$ cells $mL^{-1}$). That high density coincided, however, with relatively low cyanobacterial biovolume ($1.01 \times 10^6$ µg $m^{-3}$) indicating predominance of small cells. Maximal cyanobacteria biovolume was recorded from the surface waters of Ortigosa Reservoir in 2017 ($13.16 \times 10^6$ µg $m^{-3}$), coinciding with total cyanobacterial cell number of $2.3 \times 10^5$ cells $mL^{-1}$). Chlorophyll *a* concentration also was significantly positively correlated with

cyanobacteria biovolume (Figure 5). PC concentrations were positively correlated with total cyanobacteria biovolume (Figure 5).

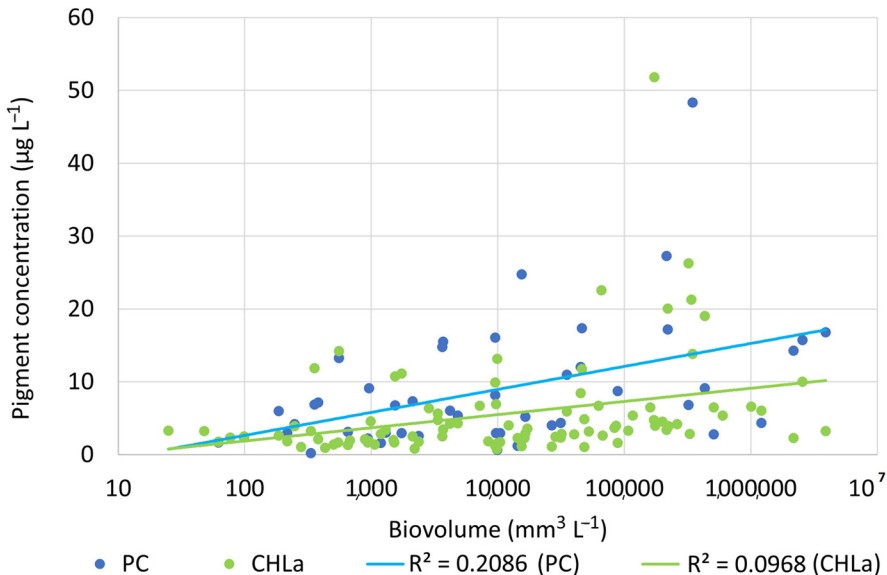

**Figure 5.** Regression of cyanobacteria biovolume versus total phytoplankton biomass as chlorophyll *a* concentration (chla) and phycocyanin concentration.

### 3.5. Pearson's Correlation and PCA

Pearson's correlations between pairs of variables are shown in Table 3. The levels of significance obtained are indicated using a colored scale. Total cyanobacteria cell number was significantly correlated with temperature, pH, nitrate, TN, TP, and total cyanobacteria biovolume. Cyanobacteria biovolume was also positively correlated with both TP and TN, whereas there was a significant negative correlation between cyanobacteria biovolume and both silica and Secchi depth.

**Table 3.** Pearson's correlation of the transformed variables related to total cyanobacteria cell number and biovolume. Colors indicate level of significance: orange, $p < 0.05$; yellow, $p < 0.01$; green, $p < 0.001$. SRP: soluble reactive phosphorus; RT: residence time.

| | Depth | Temp. | pH | Nitrate | Ammonia | Total N | SRP | Total P | Silica | RT | Biovolume | Cell number |
|---|---|---|---|---|---|---|---|---|---|---|---|---|
| **Depth** | 1 | | | | | | | | | | | |
| **Temperature** | −0.268 | 1 | | | | | | | | | | |
| **pH** | 0.001 | 0.280 | 1 | | | | | | | | | |
| **Nitrate** | −0.075 | 0.328 | −0.034 | 1 | | | | | | | | |
| **Ammonia** | −0.131 | 0.178 | −0.016 | 0.314 | 1 | | | | | | | |
| **Total N** | −0.169 | 0.451 | 0.078 | 0.887 | 0.396 | 1 | | | | | | |
| **SRP** | −0.161 | 0.148 | −0.033 | 0.304 | 0.313 | 0.346 | 1 | | | | | |
| **Total P** | −0.378 | 0.284 | 0.168 | 0.255 | 0.424 | 0.435 | 0.515 | 1 | | | | |
| **Silica** | −0.095 | −0.001 | −0.173 | 0.140 | 0.261 | 0.147 | 0.235 | 0.220 | 1 | | | |
| **RT** | 0.035 | 0.072 | −0.012 | −0.208 | 0.045 | 0.001 | −0.149 | −0.119 | −0.068 | 1 | | |
| **Biovolume** | −0.237 | 0.129 | 0.164 | 0.063 | 0.080 | 0.242 | 0.060 | 0.303 | −0.249 | 0.119 | 1 | |
| **Cell number** | −0.237 | 0.481 | 0.236 | 0.149 | 0.047 | 0.265 | 0.087 | 0.204 | −0.295 | 0.059 | 0.741 | 1 |

The results of the PCA multicomponent analysis are shown in Figure 6. The first two axes indicated 35% involvement for component 1, while 15% of the variance could be explained by component 2. Only samples with presence of cyanobacteria as indicated by PC concentrations were included, and pH data were removed. The factor scores indicated that component 1 was related to nutrient variables (N and P), indicating a relationship with eutrophication in the reservoirs. The opposite situation was found for water residence time and Secchi depth, indicating that reservoirs with smaller amounts of nutrients had clearer (less turbid) water.

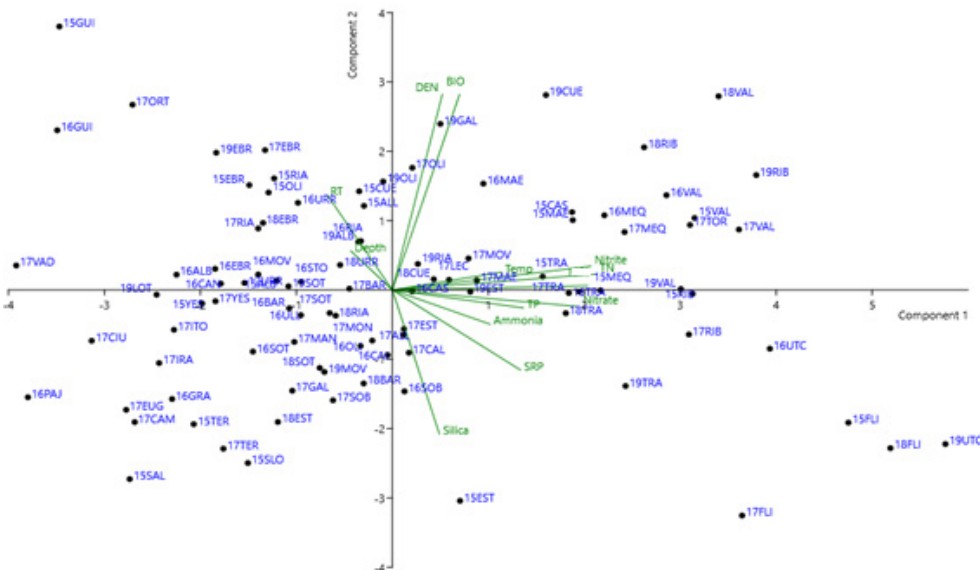

**Figure 6.** Principal component analysis of TP, TN, silica, ammonia, RT, density, biovolume, temperature. Sample sites are identified by two numbers, indicating the year, and three letters, indicating the reservoir. Additional variable acronyms are defined as follows: DEN (cyanobacterial cell number); BIO (cyanobacterial biovolume).

Component 2 was more related to the presence of cyanobacteria, and the largest factor scores were for cell number, biovolume, and water residence time, indicating that longer water residence time was more important for cyanobacteria abundance. Opposite factor scores were found for concentrations of silica and SRP, indicating reservoirs in which diatoms were more abundant than cyanobacteria. The scientific literature repeatedly recounts the importance of diatoms when the silicate concentration is not limiting their reproduction.

*3.6. PC Values Estimated from Remote Sensing*

Equation (3) was applied to 41 satellite images to estimate PC values and assess the spatial distribution of PC concentrations in the reservoirs. All of the data obtained for relative PC concentrations, both from empirical sampling and satellite measurements, are presented in Appendix A. The difference between estimated values and the actual measurements was used to identify data within the established margin of error of Equation (3), i.e., 8.1 µg L$^{-1}$, which were considered valid. After verifying the normality of data, comparison between the empirical and satellite measurements showed that 28 of the 41 samples for which satellite images were obtained were within the margin of error established for the applied algorithm. Note that even if the sample was outside that range, this would not change the reservoir classification from the WHO regarding PC concentration.

To verify the spatial distribution of [PC], examples of the application of satellite images are shown in Figure 7. Each reservoir was characterized based on the WHO classification in Table 2. In 2015, for example, Ortigosa Reservoir was at Alert level I, whereas La Loteta Reservoir was at Guidance level I. Cueva Foradada Reservoir was partly at Guidance level I and partly at Alert level II, depending on location. Barasona Reservoir, the best of the four, was at the Surveillance level. See Appendix B (Figure A1) for more examples.

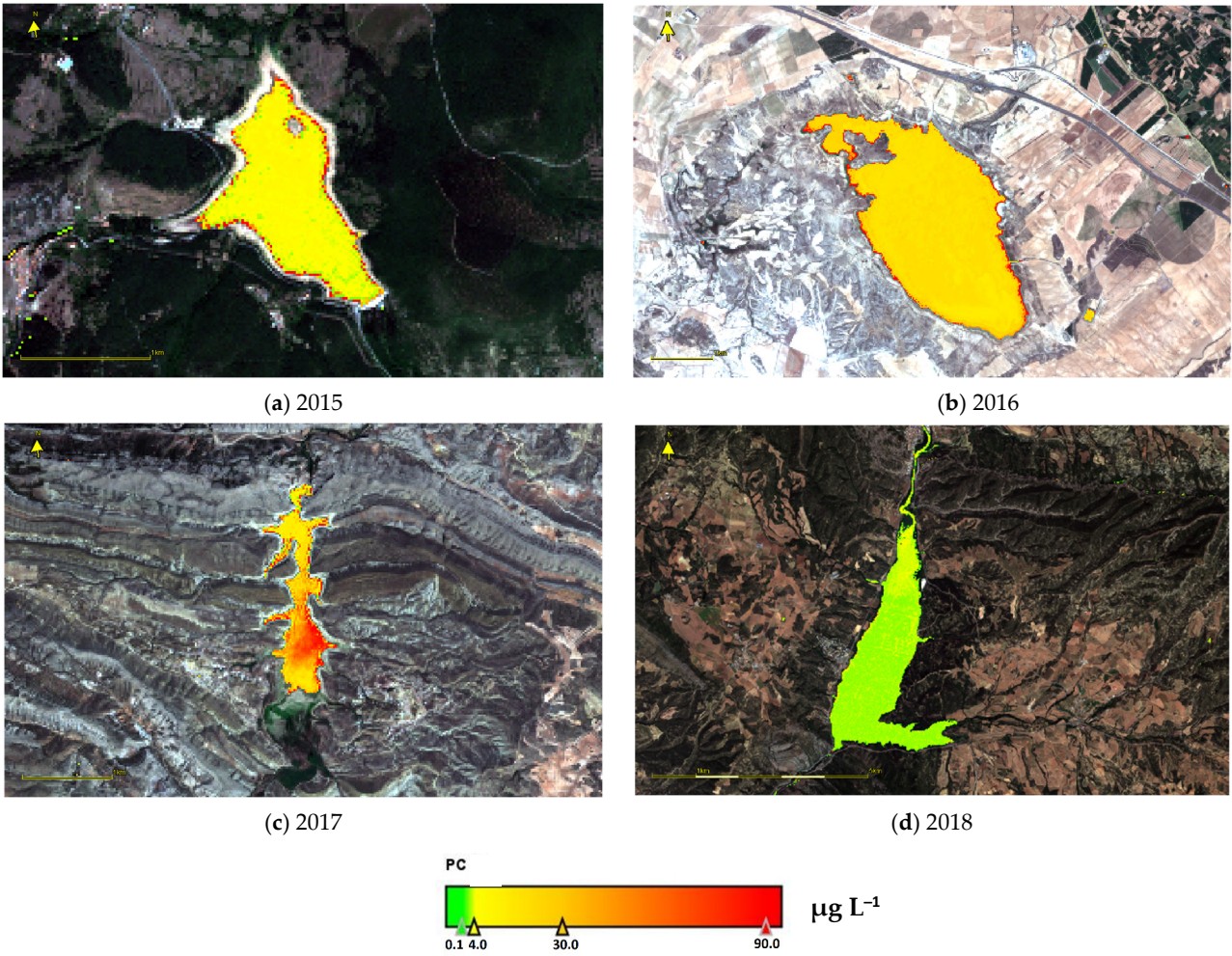

**Figure 7.** Example maps showing summer PC concentrations estimated from remote sensing of reservoirs including (**a**) Ortigosa, (**b**) La Loteta, (**c**) Cueva Foradada, and (**d**) Barasona.

## 4. Discussion

Cyanobacteria are often used as indicators of eutrophication processes in water bodies. They indicate over-enriched nutrient conditions in the ecosystem. Although reservoirs are less well studied than lakes [32], the abundance of cyanobacteria in this research was positively related to temperature, TP and TN, supporting previous research, e.g., [33]. A combination of increased anthropogenic nutrient supplies, rising temperatures, enhanced vertical stratification, increased residence time, and more extreme climatic conditions has been reported to favor cyanobacterial dominance across a wide range of aquatic ecosystems [3]. These findings support the premise that there is a direct relationship between increasing nutrients, increasing temperature, and cyanobacteria proliferation [5].

Paerl et al. [34] suggested that a combination of anthropogenic nutrient loading, rising temperatures, enhanced vertical stratification, increased residence time, and more extreme climatic conditions favors cyanobacterial dominance in a wide range of aquatic ecosystems. According to Yan et al. [35], a decreasing trend in planktonic cyanobacteria can be expected with increasing concentrations of oxidized inorganic nitrogen, especially nitrate.

Furthermore, according to Mantzouki et al. [36], one of the variables that best explains the distribution of cyanobacteria in water bodies is water surface temperature, with a $p$-value < 0.002. It was observed in lakes in the Mediterranean area that, at temperatures greater than 25 °C, the number of cyanobacteria increased abruptly. This research mainly describes cyanotoxins generated by cyanobacteria in a general manner, as increasing

temperature favors the development of cyanotoxins, rather than, as is the case with some nutrients, favoring individual toxins.

Residence times varied between the reservoirs, and in the case of longer residence times, less renewal of the water mass took place, which means that the accumulation of cyanobacteria was higher and biovolume was greater. A study [33] was carried out on the physicochemical variables that affect the proliferation of cyanobacteria in lentic and lotic waterbodies, analyzing chlorophyll *a*, total nitrogen, total phosphorus, and water temperature. Nutrient concentrations most significantly influenced the development of cyanobacteria in lotic ecosystems, while in stagnant waterbodies such as reservoirs or lakes, their effect was not so significant [33]. Even so, the abundance of cyanobacteria in lentic ecosystems was higher, which may primarily be due to a factor not taken into account, such as residence time, since this type of ecosystem has a lower turnover, and therefore cyanobacteria accumulate in greater quantities, forming algal blooms, as observed in our results.

There commonly is an inverse relationship between cyanobacteria biovolume and the content of silicates. Silicates are limiting nutrients for diatoms, and diatoms proliferate when there is high availability of dissolved silicate, indicating good water quality. According to Zhang et al. [37], the environmental factors that most significantly affect diatoms include water temperature ($p = 0.002$) and silicate concentration ($p = 0.036$), where diatoms are better adapted to the optimal temperature conditions corresponding to seasonal changes, while other environmental and physicochemical variables, such as nitrate, chlorophyll *a*, or chemical oxygen consumption, are apparently not as important. It has also been observed that, with decreasing temperature, although the concentration of silicate may remain high, diatom abundance also decreases, and some diatom species are more resistant to temperature changes than others [38].

It is difficult to understand the complex interactions between factors that control why cyanobacteria become toxic and release cyanotoxins, although this is one of the most studied aspects of cyanobacteria at present. High temperatures, high luminosity, low wind speed, and the availability of nutrients such as nitrogen and phosphorus can be important factors [33], while alkaline pH could also constitute a triggering factor [38]. We obtained the same correlations as Jiang et al. [33] for some variables such as temperature, nutrient availability (nitrogen and phosphorus), and water pH. Other variables, such as high luminosity or wind speed, were not analyzed in our study. As can be seen from the results of this article, there is a direct correlation between the tendencies of the aforementioned variables and explosions of cyanobacteria in waterbodies.

Cyanobacterial biomass (biovolume) in these reservoirs did not reach the high level of concern suggested by the WHO guidance [11]. Nevertheless, 20 reservoirs were assessed at Alert Level I, and two were at Guidance Level I. Hence, the expected continued increase in TP and TN concentrations as northeastern Spain develops in the coming decades is a concern, since warming and other aspects of climate change are already acting with nutrient enrichment in other regions to promote cyanobacterial predominance [5,35].

According to Sòria-Perpinyà et al. [39], the use of satellites for the detection of phycocyanin, and, in consequence, cyanobacteria, represents a possibility for the observation of the spatial distribution of phycocyanin, and the heterogeneity thereof, which is not possible using current field sampling techniques alone.

The increase in nutrients can only lead to water quality being seriously compromised in the next few years. Hydraulic administration can intervene appropriately in water management to use deep outflow from reservoirs to promote the renewal of the water in the hypolimnion, and surface outflow to limit algal biomass in the summer [40]. Additionally, the management of catchment plans should focus on the delimitation and encouragement of agro-ecological use zones, such that they will act as buffer zones, reducing the entry of nutrients, as well as limiting the use of chemical fertilizers in conventional agriculture [40].

## 5. Conclusions

This study supports previous work indicating that pH, water residence time, and both TP and TN can strongly influence reservoir cyanobacteria. Satellite imagery from Sentinel-2 was useful for monitoring water quality and cyanobacteria detection. It has strong potential as an early warning system to mitigate adverse impacts of CHABs on human health and the environment. The combination of Sentinel-2—or another satellite—with additional techniques, such as sampling and laboratory analyses, provides more comprehensive and accurate evaluation of the cyanobacteria presence and spatial distribution in freshwater.

CHABs are expected to be favored by human-related nutrient loading, rising temperatures, and extreme weather conditions due to climate change. Cyanobacteria proliferation, in turn, will have serious consequences for aquatic biota and humans, both indirectly and directly, because many affected waters are increasingly depended upon for potable supplies, other freshwater demands, fish for human consumption, and recreational activities. Data can be checked in supplementary materials.

**Supplementary Materials:** The following supporting information can be downloaded at: https://www.mdpi.com/article/10.3390/w15142538/s1. Sheet 1 (S1): all reservoirs classified by name through the years. Sheet 2 (S2): all reservoirs sorted by year. Physicochemical data of the reservoirs, cyanobacteria biovolume and cell density are shown at S1 and S2. Sheet 3 (S3): Coefficient results for each variable to calculate the PCA can be seen at S3.

**Author Contributions:** Conceptualization, J.S. and X.S.-P.; methodology, R.P.-G.; formal analysis, X.S.-P.; investigation, E.V., J.S., M.D.S. and X.S.-P.; data curation, X.S.-P.; writing—original draft preparation, R.P.-G.; writing—review and editing, J.S. and X.S.-P.; supervision, E.V.; project administration, E.V.; funding acquisition, E.V. and J.S. All authors have read and agreed to the published version of the manuscript.

**Funding:** This research was funded by Ebro Basin Authority contracts to E.V. and J.S. years 2015 to 2019. The APC was funded by MDPI.

**Institutional Review Board Statement:** Not applicable.

**Informed Consent Statement:** Not applicable.

**Data Availability Statement:** Data are free available at http://chebro.es (accessed on 15 February 2023).

**Acknowledgments:** Ebro Basin Authority.

**Conflicts of Interest:** The authors declare no conflict of interest.

## Appendix A

**Table A1.** List of reservoirs, indicating abbreviation and PC concentration results, both measured in situ and estimated. PC concentrations determined by remote sensing.

| Reservoir and Year | Abbreviation | PC ($\mu$g L$^{-1}$) In Situ | PC ($\mu$g L$^{-1}$) Satellite | RMSE $\mu$g/L | Reservoir and Year | Abbreviation | PC ($\mu$g L$^{-1}$) In Situ | PC ($\mu$g L$^{-1}$) Satellite | RMSE $\mu$g/L |
|---|---|---|---|---|---|---|---|---|---|
| Canelles 2016 | 16CAN | 1.63 | 3.86 | 1.57 | Ebro 2018 | 18EBR | 14.79 | - | - |
| La Sotonera 2016 | 16SOT | 3.00 | 5.47 | 1.74 | Lechago 2018 | 18LEC | 4.31 | 5.22 | 17.43 |
| La Tranquera 2016 | 16TRA | 10.94 | 5 | 4.20 | Monteagudo 2018 | 18MON | 8.70 | 4.39 | 3.05 |
| Mansilla 2016 | 16MAN | 1.54 | 6.65 | 3.61 | Urrunaga 2018 | 18URR | 16.79 | 4.8 | 9.47 |
| Santolea 2016 | 16STO | 4.01 | 4.01 | 0 | El Val 2018 | 18VAL | 11.70 | - | - |

**Table A1.** *Cont.*

| Reservoir and Year | Abbreviation | PC (µg L⁻¹) In Situ | PC (µg L⁻¹) Satellite | RMSE µg/L | Reservoir and Year | Abbreviation | PC (µg L⁻¹) In Situ | PC (µg L⁻¹) Satellite | RMSE µg/L |
|---|---|---|---|---|---|---|---|---|---|
| Ullibari-Gamboa 2016 | 16ULL | 1.24 | - | - | Oliana 2018 | 18OLI | 9.05 | 4.11 | 3.49 |
| Sobrón 2016 | 16SOB | 2.95 | 3.85 | 0.63 | Sobrón 2018 | 18SOB | 15.69 | 4.4 | 8.98 |
| Alloz 2017 | 17ALL | 3.11 | 4.38 | 0.89 | Terradets 2018 | 18TER | 24.77 | 10.41 | 12.15 |
| Ebro 2017 | 17 EBR | 4.34 | - | - | Cueva Foradada 2018 | 18CUE | 13.24 | 12.26 | 0.69 |
| Eugui 2017 | 17EUG | 2.93 | 3.66 | 0.52 | Mezalocha 2018 | 18MEZ | 5.96 | 8.33 | 1.67 |
| Irabia 2017 | 17IRA | 0.21 | - | - | La Sotonera 2018 | 18SOT | 15.52 | 5.65 | 6.98 |
| Itoiz 2017 | 17ITO | 2.37 | 4.61 | 1.58 | Barasona 2018 | 18BAR | 2.57 | 4.13 | 1.11 |
| Maidevera 2017 | 17MAE | 2.94 | - | - | Rialb 2018 | 18RIA | 5.19 | 4.4 | 0.56 |
| El Val 2017 | 17VAL | 6.80 | - | - | La Tranquera 2018 | 18TRA | 17.35 | 4 | 9.44 |
| Oliana 2017 | 17OLI | 15.71 | 5.32 | 9.35 | Flix 2018 | 18FLI | 14.29 | 7.64 | 7.74 |
| La Peña 2017 | 17PEÑ | 4.21 | 5.02 | 0.57 | Ribarroja 2018 | 18RIB | 17.19 | 8.75 | 8.86 |
| Terradets 2017 | 17TER | 7.15 | 14.94 | 8.51 | Ebro 2019 | 19EBR | 4.38 | 2.86 | 1.07 |
| Yesa 2017 | 17YES | 2.23 | 3.87 | 1.16 | Oliana 2019 | 19OLI | 2.76 | 2.86 | 0.07 |
| Cueva Foradada 2017 | 17CUE | 3.70 | 12.05 | 9.52 | Sobron 2019 | 19SOB | 11.61 | 3.85 | 8.48 |
| Gallipuén 2017 | 17GAL | 4.21 | - | - | Estanca de Alcañiz 2019 | 19EST | 8.18 | - | - |
| Mezalocha 2017 | 17MEZ | 34.08 | 19.77 | 10.13 | Gallipuen 2019 | 19GAL | 27.28 | 17.17 | 7.15 |
| Moneva 2017 | 17MOV | 48.34 | - | - | La Loteta 2019 | 19LOT | 9.11 | 9.98 | 0.62 |
| Las Torcas 2017 | 17TOR | 2.98 | 4.4 | 1.01 | Moneva 2019 | 19MOV | 6.89 | 28.1 | 14.99 |
| Camarasa 2017 | 17CAM | 1.66 | - | - | La Sotonera 2019 | 19SOT | 7.33 | 5.65 | 1.19 |
| Rialb 2017 | 17RIA | - | - | - | Utchesa-Seca 2019 | 19UTC | 16.09 | 2.83 | 9.37 |

**Table A2.** Reservoir classifications according to the Water Framework Directive 2000/60/EC [13].

| Type | Reservoirs | Mixing Regime | Geology | Humidity Index (HI) | Basin Area | Annual Temperature |
|------|-----------|---------------|---------|---------------------|------------|--------------------|
| 1 | Lanuza, Pajares. | | Siliceous (alkalinity < 1 meq/L) | HI > 0.74 Medium Humidity | Header and upper reaches (basin area < 1000 km$^2$) | <15 °C |
| 7 | Albiña, Alloz, Búbal, Ebro, El Val, Escales, Escarra, Eugui, Irabia, Itoiz, Lechago, Maidevera, Mansilla, Monteagudo, Ortigosa, Sopeira, Ullívarri-Gamboa, Urdalur, Urrúnaga, Vadiello. | Monomictics | Calcareous Alkalinity > 1 meq/L) | HI > 0.74 Medium Humidity | Header and upper reaches (basin area < 1000 km$^2$) | <15 °C |
| 9 | La Peña, Mediano, Oliana, Sobrón, Terradets, Yesa. | | | | (basin area > 1000 km$^2$) | |
| 10 | Ciurana, Cueva Foradada, Gallipuen, Guiamets, La Estanca, La Loteta, La Sotonera, Las Torcas, Margalef, Mezalocha, Moneva, Pena, Utchesa-Seca | | | HI < 0.74 Low Humidity | Header and upper reaches (basin area < 1000 km$^2$) | |
| 11 | Balaguer, Barasona, Calanda, Camarasa, Canelles, Grado, La Tranquera, Rialb, San Lorenzo, Santolea, Talarn. | | | | (basin area > 1000 and < 25,000 km$^2$) | |
| 12 | Caspe, Flix, Mequinenza, Ribarroja. | | | | Lower sections of the main axes (basin area > 25,000 km$^2$) | |
| 13 | Cavallers, Llauset. | **Dimictics** | | HI > 2 High Humidity | | |

## Appendix B

Examples of satellite imagery of PC for reservoirs in the Ebro Basin. Cavallers, Llauset and Sallente reservoirs are located in the Pyrenees according to WHO classification (Appendix A, Table A1). They are considered oligotrophic waters with low levels of nutrient enrichment. Other reservoirs, such as Terradets and Canelles, are also near the Pyrenees, but are subject to nutrient pollution from human influence. Their ecological status is worsening.

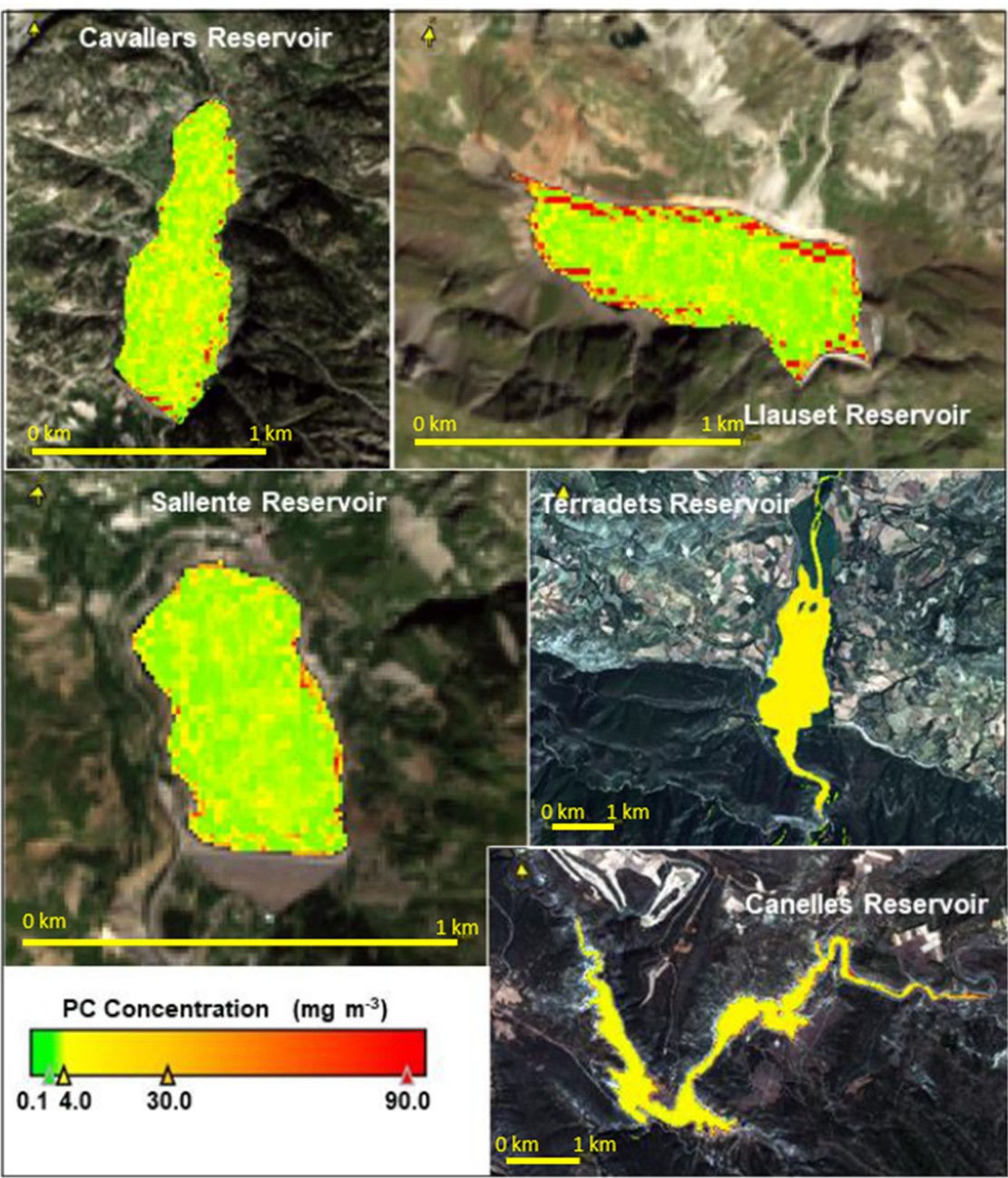

**Figure A1.** PC in five reservoirs from the Ebro basin according to WHO classification (Table 2).

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
