# Peer review of "Relationship between Cyanobacterial Abundance and Physicochemical Variables in the Ebro Basin Reservoirs (Spain)"

_water, doi:10.3390/w15142538_

Round 1
Reviewer 1 Report
The manuscript submitted by Pérez-González and colleagues focused on the analysis of physicochemical factors that could impact proliferation and biovolume of cyanobacteria. In general, this is another study with comparatively comprehensive data supporting that the bloom of cyanobacteria is correlated to the environmental parameters, such as pH, temperature, nutrients, etc. This study has further indicated the feasibility of using satellite detection of phycocyanin as assistant approach in the measurement of spatial distribution of cyanobacteria. However, the novelty of the study must be clearly justified as also described in the discussion, many other studies have already present the similar findings.
Comments:
1. In the abstract, it seems that the authors wanted to emphasize the bloom of cyanobacteria will impact water quality and thus affect water supply. However, the logic between the first two sentences and the following context should be further justified.
2. Line 26, “The results show alarming levels of cyanobacteria in a large part of the reservoirs.” This is not in line with the description in the discussion (Line 537).
3. The introduction of the manuscript needs to be carefully revised. The logics between paragraphs are weak and much information are not tightly relevant to the scope of the study.
4. In the results, it is mentioned that there are 58 samples collected, however, there are 50 samples included in the table A1.
5. Line 286, it should not be described as “quite similar” if the temperatures are ranged from 15 to 25.
6. The authors should introduce the classification of different types of reservoirs since the concept of types is important throughout the manuscript. Table A2 needs to be revised to highly types and reservoirs.
7. The authors should present all the data, including residence times, pH, nutrients, ect., preferably in supplementary document, rather than only presenting the boxplots.
8. What is the difference between Dots and stars in the figures?
9. The paragraph 1 in the discussion should be removed if there is no specific reason to include it.
10. Line 530-536, it is hard to understand the relevance between the cited paper and the description in the last sentence about the results from the current manuscript.
11. Link in Reference 20 cannot direct to any webpage, should be updated.
12. Line 320, “OECD” should be defined.
13. Typos: Line, 40, “This” should be “The”; line 41, “causes” should be “caused”; line 44, “noy” should be “not”; line 162, “outlet” should be “outflow”.
Some typos are in the manuscript.
Author Response
The responses to reviewer 1 are attached in the PDF below.

Reviewer 2 Report
Journal: Water (ISSN 2073-4441)
Manuscript ID: water-2346945
Type: Article
Title: Relationship between cyanobacterial biovolume and physicochemical variables in some Spanish reservoirs.
Section: Biodiversity and Functionality of Aquatic Ecosystems
Special Issue: Eutrophication and Harmful Algae in Aquatic Ecosystems
This manuscript aims to analyse the concentrations and densities of cyanobacteria in reservoirs in the Ebro basin (Spain) and the variables that influence the proliferation of cyanobacteria, such as temperature, pH, and residence time, concentrations of soluble phosphorus, total phosphorus, nitrates, nitrites, inorganic nitrogen and total nitrogen. However, there are some issues need to be addressed. The presentation is not good. Some detailed comments and suggestions are listed as follows:
1. Title
Please change to “Relationship between cyanobacterial biovolume and physicochemical variables in some Spanish reservoirs in the Ebro basin, Spain”.
2. Abstract
Lines 10-14
“The need for an increasing water supply as a result of massive population growth is a continuous cause of problems. Subjecting the water supply to increased pressure will diminish its quality, affecting the flora, fauna and even human beings that depend on the water, and thus necessitating either an increase in the cost of water treatment, or the loss of water bodies for human consumption.”
These sentences are too general. Please delete them.
3. More results of this study should be presented in the abstract.
4. Lines 16-20
“This study focused on the different physicochemical variables impacting inland water bodies in the NE region of the Iberian Peninsula, in the Ebro River basin, including temperature, residence time, pH, total nitrogen, inorganic nitrogen, total phosphorus, orthophosphate, silicate and phycocyanin, in order to establish the relationship of these variables with biovolume of cyanobacteria, as well as their effect on proliferation.”
This sentence is too long. Please dive it to two sentences or more. What is NE? Please spell the full them when it first occurs.
5. Section “1. Introduction”
Lines 33-38
“Nowadays, inland water bodies are subject to significant pressure, mainly as a result of human activities. On the one hand, there is a growing need for water supplies due to population growth, while on the other hand, there is a continuous risk of surcharges into these waters occurring, in most cases without prior treatment. As a consequence, the ecological status of water bodies is worsening, resulting in the increasing occurrence of algal blooms, presenting a great risk to the flora and fauna of the lakes, as well as to human beings.”
Please insert the references.
6. Lines 70-71
“All these factors make cyanobacteria on of the most important primary producers within the food chain as well as a resource [4].”
Please change “on” to “one”.
7. Lines 72-73
“Cyanobacteria are able to fix atmospheric N2, but they are also able to obtain it from water through the assimilation of NH4+, NO2- or dissolved inorganic nitrogen (NIO).”
Please change “it” to nitrogen. What do you mean by dissolved inorganic nitrogen (NIO)?
8. Lines 91-95
“The risk to human health mainly arises through potential exposure to these toxins, either by direct contact with contaminated water or by ingestion of contaminated water due to the dependence of population on accessibility of water for regular consumption, while microalgae do not necessarily form scum or foam that can be identified visually in water [13].”
Cyanotoxins can also enter human bodies through the food chain/web. Please read and cite the following paper.
Chen et al., 2021. Challenges of using blooms of Microcystis spp. in animal feeds: A comprehensive review of nutritional, toxicological and microbial health evaluation. https://doi.org/10.1016/j.scitotenv.2020.142319
9. Table 1
What is PC? Please spell the full them when it first occurs.
10. Section “2. Materials and Methods”
Figure 1
Please present location of Ebro basin in Spain. Also, longitude and latitude should be presented in the map. Which reservoirs did you study in this research? Please mark the studied reservoirs in the figure.
11. Lines 147-148
“Samples were collected from different reservoirs in the summer period (June to September) in the Ebro River basin from 2015 to 2019.”
Line 270
“The reservoirs considered in this study were sampled between 2015 and 2019.”
In the Appendix A Table A1, it seems that no samples were collected in 2015. Please check it.
12. Section “2.2. In situ sampling”
How did you collect the water sample? From the surface water or from the bottom? Please add the details in the revised manuscript.
13. How many sampling points in each reservoir? Please present number of sampling points in each reservoir and information of longitude and latitude in the revised manuscript.
14. Section “3. Results”
Table A2
What do types 1, 7, 9, 10, 11, 12, and 13 mean? Please present some details in the revised manuscript. Otherwise, your readers will have to go to previous paper (Water Framework Directive 2000/60/EC) back and forth when reading this paper.
15. Figure 2a
Please change letter of reservoir 12 from “df” to “de”. Please check the results of statistical analyses.
16. Figure 2b
Please change letters of reservoirs 7, 9 and 10 to “a”. Please check the results of statistical analyses.
17. Lines 305-307
“The basic pH of the water benefits the proliferation of cyanobacteria; therefore, the higher the pH, the higher the concentration of cyanobacteria.”
Results of cyanobacteria are not presented here. Please remove this sentence or move it to other place.
18. Lines 336-339
“The data processed over the course of five years evidence a slight increase year after year, with the maximum peak of almost 4.79 mg L-1 being observed in 2017 in El Val reservoir, corresponding to a high biovolume of cyanobacteria of 321,974 µg m−3.”
Results over five years are not presented here. Please remove this sentence or move it to other place.
19. Lines 341-345
“In the case of soluble inorganic nitrogen (NIO), the correlation with the biovolume of cyanobacteria was negative and not significant, so greater amounts of dissolved inorganic nitrogen in the water did not greatly affect the concentration of cyanobacteria. This is because cyanobacteria do not assimilate it in the same way as other types of nitrogen-group compounds.”
Results are not presented here. Please remove these sentences or move them to other place.
20. Figure 5b
Please change letters of reservoirs 9, 10 and 12 to “ab”, “ac” and “c”, respectively. Please check the results of statistical analyses.
21. Lines 355-358
“In reservoirs in which there were greater amounts of silicates, there were fewer cyanobacteria, as in the case of Lechago in 2015, where the concentration of silicate peaked at around 16.24 mg L-1, while zero cyanobacteria were registered.”
Results of cyanobacteria are not presented here. Please remove this sentence or move it to other place.
22. Figure 9
Please change letters of reservoirs 7 and 10 to “a”. Please check the results of statistical analyses.
23. Figures 2-7, 9
Please present results of each reservoir in the supplementary file. Then readers can better understand your paper.
minor issues
Author Response
The responses to reviewer 2 are attached in the PDF below.

Round 2
Reviewer 1 Report
The manuscript is significantly improved after revision. I would support its publication in the Journal.
Author Response
Thank you for your corrections in the first review, they were a great help for improving our work. We have made some additional corrections as recommended by the other reviewer, like some changes in the figures in the results section.
Once again, thank you for recommending our article for publication in Water.
Reviewer 2 Report
Journal: Water (ISSN 2073-4441)
Manuscript ID: water-2346945-peer-review-v2
Type: Article
Title: Relationship between cyanobacterial biovolume and physicochemical variables in the Ebro basin reservoirs (Spain)
Section: Biodiversity and Functionality of Aquatic Ecosystems
Special Issue: Eutrophication and Harmful Algae in Aquatic Ecosystems
This manuscript aims to analyse the concentrations and densities of cyanobacteria in reservoirs in the Ebro basin (Spain) and the variables that influence the proliferation of cyanobacteria, such as temperature, pH, and residence time, concentrations of soluble phosphorus, total phosphorus, nitrates, nitrites, inorganic nitrogen and total nitrogen. However, there are some issues need to be addressed. I have pointed out several errors, but the authors did not make any changes. In fact, the reviewer only helps authors to improve the quality of the manuscript. It is the authors’ responsibility to present their best work to the readers.
1. The authors did not understand how to use different letters to show significant differences among groups. For example, "a" and "b" or "bc" or "c" have significant differences, but "b" and "ab" or "bc" have no significant differences. I have pointed out several errors, but the authors did not make any changes. For example,
1) Figure 2a
If the results of statistical analyses presented are right, please change letter of reservoir 12 from “df” to “de”. Please check the results of statistical analyses.
2) Figure 2b
If the results of statistical analyses presented are right, please change letters of reservoirs 7, 9 and 10 to “a”. Please check the results of statistical analyses.
3) Figure 5b
If the results of statistical analyses presented are right, please change letters of reservoirs 9, 10 and 12 to “ab”, “ac” and “c”, respectively. Please check the results of statistical analyses.
4) Figure 9
If the results of statistical analyses presented are right, please change letters of reservoirs 7 and 10 to “a”. Please check the results of statistical analyses.
2. Please read and carefully check through all the manuscript text, tables and figures. It is the authors’ responsibility to present their best work to the readers.
ok
Author Response
First of all, thank you very much for your comments and revisions. As you may have noticed we have indeed taken many of your corrections that you made in the first review, and, we had submitted an excel with supplementary material with all the results in the previous submission.
Regarding to the statistical results, we have changed figures 2a and 2b as you suggested. For 5b, we have changed some of the letters, but not all of them as you indicated, and in Figure 9 we have left it unchanged. I will explain you why we didn’t change them all; we think that our statistical results were fine, for this we have attached a word document with the ANOVA statistical test, so you can see the results by yourself. The parts marked in yellow are the ones that have been changed, but in some cases, the difference between certain letters was so small and in consequence, not significant, and this is because we left them as they were.